# Barriers to the Development of Organic Farming: A Polish Case Study

**Władysława Łuczka** [1] and **Sławomir Kalinowski** [2,*] 

[1]  Department of Economics, Poznan University of Life Sciences, 60-639 Poznan, Poland; wladyslawa.luczka@up.poznan.pl

[2]  Department of Rural Economics, Institute of Rural and Agricultural Development of the Polish Academy of Sciences, 00-330 Warsaw, Poland

*  Correspondence: skalinowski@irwirpan.waw.pl; Tel.: +48-607-994-578

**Abstract:** The main purpose of this paper is to explore farmers' opinions on the barriers to the development of organic farming. A survey was carried out with 262 Polish organic farmers in order to classify the barriers to organic farming development into production, and economic aspects, market aspects and institutional and regulatory aspects. As a next step, a detailed analysis was performed of how the farmers view these barriers. According to this study, Polish organic farmers attach greater importance to economic factors than to non-economic ones. Low yields and production volumes are the reason why many farmers see organic farming as being risky. More than 80% and nearly 60% of farmers covered by this study found the production risk to be very high or high, respectively, during and after the conversion period. Most farmers say they intend to continue their organic production activity only if financial support is provided. Nearly one in five farms (18.3%) want to discontinue organic production in future. This is especially true for two types of farming: specialized grazing livestock farms and mixed holdings. The farmers believe that market aspects and institutional and regulatory factors are the key barriers to the development of organic farming. The findings regarding the role of institutional barriers and communications from regulatory institutions, which affect the farmers' decision-making processes, are of particular importance. In Poland, the main institutional problem is the instability of laws applicable to organic farming, which adds to the farmers' uncertainty and decision-making risks. The case study of Poland, which is among the emerging markets for organic food, shows that a stable and coherent support policy is a condition for organic-farming development.

**Keywords:** organic farming; feedback from farmers; motives behind conversion; development barriers

## 1. Introduction

Ever since it emerged and started to develop, organic farming has been viewed as an alternative agricultural system which could contribute to solving a number of environmental problems and food quality issues [1–4]. As such, it has become increasingly accepted by producers, consumers and politicians [5]. The low risk of organic food contamination (especially with pesticide residues) resulting from production standards, the norms and principles of manufacturing provided for in legal regulations, and the certification system, are the main factors that contribute to a positive image of organic farming and its products. The commercial outcome of this is the growing demand for organic food, especially from consumers who attach great importance to health [6–12]. This has long reinforced the development of organic farming, as reflected in its high growth rates that have been recorded since the 1990s. However, in recent years its development (measured as growth in organic farm numbers and size) has been fluctuating in some countries [13–17].

The pace of organic farming development is disturbed by changes in its structure, which consist of the fact that some farms are discontinuing organic practices while others are adopting them [18]. In some countries, the decline in organic farm numbers exceeds the number of newcomers; this is also reflected by the reduction in the area containing organic crops. For instance, in the European Union the organic farming discontinuation rate for 2005 was 7.3% [19]. This means that organic farming faces barriers which, if identified, may play a significant role in its further development and can have a particular impact on future organic farming policy. The development of an effective policy that promotes organic farming requires the identification not only of factors that encourage the selection of organic production methods, but also of factors that lead farms to discontinue them [20].

Based on surveys [21–23] and in-depth interviews [24–27], many studies on organic farming focus on factors that lead farmers to opt for organic or conventional farming. A number of researchers have attempted to estimate the number of traditional farmers who are considering going organic [22,25]. According to an analysis of determinants of organic production methods in Finland, the decline in the prices of conventional products and the growing subsidies are important factors that encourage the transition to organic farming. Furthermore, farms with a larger area for crops and farms with lower yields are more likely to go organic [28]. In their study on Greek farmers, Geniusa et al. [29] pointed out the importance of information in the switch to organic methods. Numerous studies have identified the conversion period as the main barrier to keeping an organic farm afloat [30,31]. An important factor that discourages farmers from selecting organic production patterns is the conversion period, which requires them to rapidly restructure their farms [32]. On the one hand, this entails the need to incur many additional costs involved in investments and access to information; on the other, it has an effect on the periodic drop in incomes due to such factors as a decline in yields and the inability to earn a price premium.

The particularities of organic production also require farmers to acquire new knowledge and skills. According to Morgan and Murdoch [32], organic farming know-how plays a crucial role because organic farms no longer adhere to the productivity paradigm and, hence, can no longer rely on the related knowledge. Instead of cumulative increments in knowledge, which is typical of conventional systems, the conversion process requires farmers to forget what they have learned from their intensive production practices. On the other hand, farmers must gain new knowledge, which is not easy in the case of organic farming, as the whole system of agricultural know-how focuses on conventional farming. This is corroborated by research that has found lack of knowledge to be among the key barriers to switching to organic farming [31].

Over recent years, some studies have focused on identifying the reasons for discontinuing organic farming. The classifications of these motives differ greatly and the boundaries between some motives are blurred. Rigby et al. [33] defined four groups of reasons for discontinuing organic farming, namely those related to (1) market and sales, (2) costs, (3) agronomic problems, and (4) personal motives. They found a relationship between the decision to discontinue organic farming and the farmers' demographics. Older, educated and female farmers are more inclined to return to traditional farming. Darnhofer et al. [27] found farmers to be heterogeneous in their attitudes and goals, and claimed this had an effect on their choice of farming methods. They identified five types of farmers based on business strategy and values, as follows: "committed conventional farmers", "pragmatic conventional farmers", "environmentally-aware yet non-organic farmers", "committed organic farmers" and "pragmatic organic farmers". They determined that new organic farmers differ from their predecessors (pioneers) in that they are less environmentally committed and are business-focused. This is also confirmed by other studies [34–36].

Some studies suggest that farmers who work on a part-time basis or are engaged in professional activity are more inclined to discontinue organic farming [18,37,38]. The investigation into the relationship between the decision to discontinue organic farming, the cropping mix and animal husbandry proved that the likelihood of discontinuation is low in cattle farms and in farms with a larger share of land under vegetables [38]. It is the opposite for milk farms and bovine meat producers,

who exhibit an above-average likelihood of quitting the organic sector. Farms that keep pigs or poultry for fattening are more likely to return to conventional methods. This is also true for farms with a larger share of pastureland and cereals.

The importance of subsidies in the decision for organic farming was confirmed in a Finnish study [28] and in a comparative analysis illustrated by the example of EU countries [39]. Flaten et al. [40] proved that economic and regulatory issues were the main reasons why Norwegian farmers discontinued organic practices. One of the ways to encourage farmers to maintain their organic farming system is to focus the support mechanism more precisely than ever before and to reduce the frequency of amendments to regulations for organic production standards. In a study by Sahm et al. [41], farmers' decisions to discontinue organic production were impacted by different factors, such as economic aspects, matters related to certification and production techniques, and macro-environmental issues. However, the study confirmed the key importance of economic factors.

Most Polish studies on organic farming have focused on analyzing the condition of organic farming, selected aspects of efficiency, and development outlooks and conditions [42–49]. These studies mostly highlighted the existence of favorable economic and natural conditions for the development of organic production methods in Poland. As regards the assessment of organic farming's development capacity, the prevailing opinion is that development opportunities are greater than factors threatening the development trend [43,50].

In Poland, the barriers to organic farming development are a problem addressed in only a few studies. Both the barriers and the scale of their impact on organic farms are rarely identified. No studies have been carried out with farms that have discontinued organic production, because they are not recorded in any statistical system. Recent years have witnessed the emergence of one paper which partly addressed but failed to exhaust the problem of barriers to organic farming development [51]. According to this study, the lack of markets, difficulties in selling their products and limited demand were the key barriers identified by most farmers. Only a negligible percentage of the interviewees cited other obstacles, such as high production costs, large labor inputs and poor profitability. This may mean that Polish farmers believe market barriers are more important than production barriers. These studies form part of a research approach which mostly sees demand-side and distribution-related factors as the barriers to organic farming development, while attributing a lesser role to production aspects.

The purpose of this study was to explore the opinions of organic farmers on the barriers to the development of organic farming. This study classified the barriers to organic farming development into production and economic aspects, market aspects and institutional and regulatory aspects. As a next step, a detailed analysis was performed of how the farmers view these barriers. That aspect of organic farming development has not yet been researched on in Poland.

In 2004, Poland acceded to the EU; that year also marked a breakthrough in the development of Polish organic farming. Earlier, organic farming was of minor importance (with 2286 organic farms in 2003, and a total of 2928 thousand farms). Since 2004, Poland has been recording rapid development of organic farming, which is mostly related to the dedicated support available under agri–environmental programs [52–56]. Moreover, the development of organic farming has been stimulated by consumer demand for high-quality food. In 2018, the Polish market for organic food was worth ca. EUR 245 million, i.e., 0.3% of the value of the food market. Over recent years, the organic food market has grown at an annual rate of 10–20%, with imports being an important contributor. A large part of organic raw materials are exported to other EU countries (France, Germany) and third countries due to greater price premiums, and are subsequently imported as processed products. The annual spending of a Polish consumer on organic food is ca. EUR 6, i.e., less than one tenth of what EU consumers spend on average. There is not enough commitment from policymakers to making consumers more environmentally aware. Since Poland joined the EU, the Ministry of Agriculture and Rural Development has participated twice in countrywide information and promotional campaigns for organic farming (in 2006 and 2012).

The authors of this paper divide the post-accession development period of organic farming into two stages. The first, ten-year stage (2004–2013) witnessed a dynamic increase in the area of organic farmland and in the number of organic farms. This was a time of quantitative changes in organic farming—between 2004 and 2013 the area of organic farmland increased by over 710%, from 82,700 ha to 670,000 ha, and the number of organic farms increased by 620% (from 3700 to 26,600). In 2013, Poland had the fifth largest area of organic farmland and the third largest number of organic farms in the EU. During that period, organic farmers were financially supported with funds available under two Rural Development Programmes (2004–2006 RDP and 2007–2014 RDP), as allocated by the European Agricultural Fund for Rural Development (EAFRD).

At that time, Poland recorded low levels of organic production, which can be partly explained by the principles for granting financial support applicable until 2013. In the initial years that followed Poland's accession to the EU, the requirements of eligibility for organic farming support under agri–environmental programs were quite liberal. Under the first and second (2004–2006 and 2007–2013) RDP, farmers received payments for environmental measures while not being required to deliver organic food to the market. As such, the farmer–market relationship was weak and the increase in the area of organic land failed to result in a commensurate growth in supply of organic food. In that period, the farmers were highly interested in horticultural crops, which earned them the highest payment rates without the need to harvest fruit during the implementation of the organic obligation.

By 2014, most Polish organic farms had discontinued livestock production. As a consequence, in that period, animal numbers grew more slowly than agricultural land [57]. The examples include larger farms, which often sought subsidies rather than embarking on a sustainable organic development path. While having a larger forage area, they either demonstrate a low animal density or totally discontinue the rearing of animals. They also increase the area of land under crops more slowly than the general rate of agricultural land growth. The idea behind organic farming is to combine plant and livestock production to enable self-sufficiency and sustainable feed and fertilizer management. This is why animal husbandry should be an integral part of organic farms. Failure to comply with this principle entails specific environmental and market consequences. First, the absence of livestock production on organic farms contradicts the concept of environmental sustainability. Second, it contributes to reducing the farms' commercial capacity and the supply of animal products to the market.

The ten-year period from 2004 to 2013 made very little contribution to making organic farms more sustainable, i.e., improving the capacity to develop in addition to ongoing operations. This period was marked by the existence of a large group of new farmers, referred to as subsidy consumers, who embarked on the organic path guided by financial incentives [53]. They were poorly linked to the organic food market. According to studies, the average rate of marketable production for organic farms was 42.6% [58]. This means it was nearly 30 percentage points below what was recorded in conventional farms (ca. 70%). The share of organic farms which are either non-commercial or have a marketable production rate of up to 20% was 30%, which means that the production business of every third farm was run on a non-commercial basis.

In the second stage of organic farming development—which started in 2014 and is still continuing—the area of organic farmland has declined every year. The net decline in organic farmland between 2018 and 2013 was 185,300 ha (−27.7%), and the net decline in the number of organic farms was 7300 (−27.4%). The beginning of this period coincided with the launch of the new support program financed under the 2014–2020 RDP, which introduced new requirements for farmers receiving organic payments. Since 2014, in the case of agricultural crops, vegetables, herbs and horticulture, the condition for granting payments under the "organic farming" measure has been the delivery of no less than 30% of harvests for processing, to other farms or for sale. For forage crops and permanent pasture, the harvests should be fed to animals or be delivered for sale or to other farms. Since 2014, greater emphasis has been placed on coupling vegetable production with livestock production. In the case of forage crops and permanent pasture, a requirement was imposed to keep a minimum number

of animals. Farmers are eligible for payments if they meet the minimum animal density of 0.3 livestock units per hectare.

As a consequence of these changes, some farmers (mostly those not engaged in production) decided to discontinue organic production and not to participate in the agri–environmental program financed under the 2014–2020 RDP. Meanwhile, the number of new organic farmers has not grown. In the second period of organic farming development, Poland has been witnessing a decline of organic farming at a rate unseen in the EU. A negative slope of the identified polynomial trend reveals a negative development trend. The area of organic farms can be reasonably expected to continue to decline in the coming years (Figure 1).

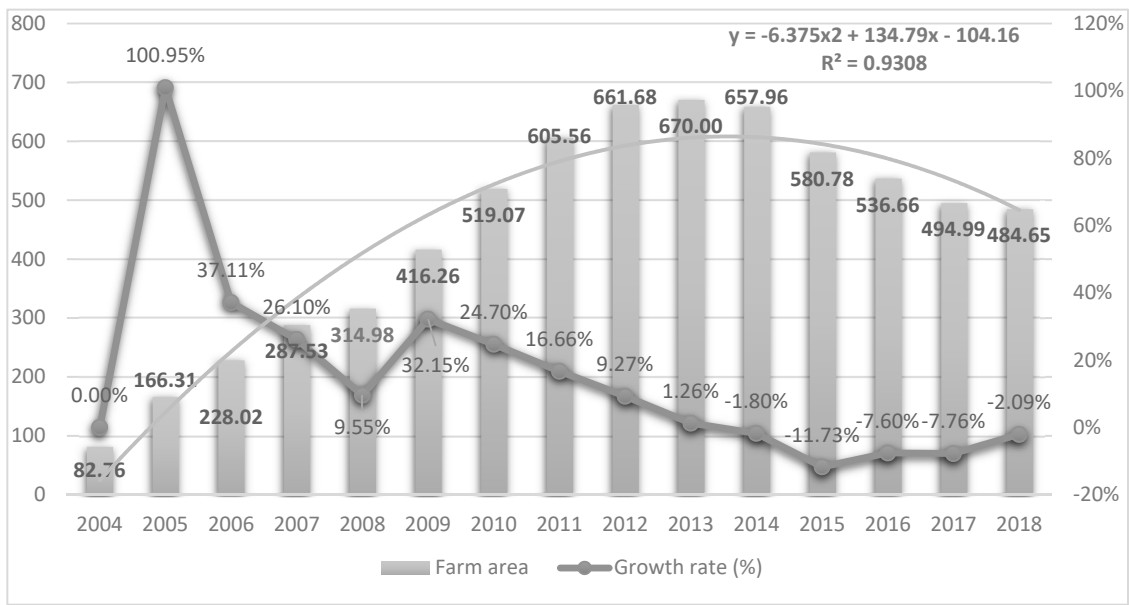

**Figure 1.** Growth and decline in the area of organic farmland in Poland between 2004 and 2018. Source: own study based on [59,60].

## 2. Materials and Methods

A two-stage survey procedure was carried out to explore farmers' opinions on the barriers to the development of the Polish organic farming sector. The first stage, designed to identify the progress in research on barriers to the development of organic farming, reviewed the relevant literature and analyzed the reports and expert assessments from the Ministry of Agriculture and Rural Development and the Inspectorate of Marketable Quality of Agri-food Products (GIJHARS), as well as data from the Research Institute of Organic Agriculture (FiBL) and the International Federation of Organic Agriculture Movements (IFOAM). The second stage was a study based on a standardized survey questionnaire for Polish organic farmers. The assumption was made that the existing farms also include ones which are considering discontinuing or restricting their organic production activity. Thus farmers' opinions on the barriers to organic farming may help to identify some policy instruments that would prevent the Polish organic farming sector from further decline.

The survey was carried out in 2019 as a CATI (computer assisted telephone interview), based on the concepts and questionnaire forms prepared by the authors. The study sample was composed of 262 farms (upon verification of survey questionnaires) from a total population of 15,470 certified organic farms (as of 2017). The farms complied with the Regulation (EC) No. 834/2007 of 28 June 2007 on the organic production and labeling of organic products. The sample used in this study was representative of the general population of certified organic farms. The farms represented all voivodeships in accordance with the territorial distribution of Polish organic farms.

The sample size formula for a finite population was used to calculate the size of the sample of organic agricultural producers covered by this study:

$$n = \frac{P(1-P)}{\frac{e^2}{Z_\alpha^2} + \frac{P(1-P)}{N}} \tag{1}$$

where $n$ is sample size, e is permissible error, $N$ is population size, and $Z_\alpha$ is value resulting from the confidence interval used; for a 95% confidence level, $Z_\alpha = 1.96$, and $P$ is the estimated proportion in the population (usually, it is set at $P = 50\%$). $P$ is the estimated expected proportion in the population covered by the study. As the proportion in the population of operators is unknown, the least favorable assumption was made, namely that $P = 50\%$, because at that $P$ level the product $P(1 - P)$ reaches the maximum value.

It is thus assumed that:

- $P = 50\%$;
- confidence level = 95%;
- permissible error = 6%;
- $N = 15{,}470$ (the number of organic farms in 2017).

The minimum sample size with this data is 262. The stratified random sampling method was used to ensure representativeness across the entire population of organic producers active in Poland. The stratification procedure took account of the number of organic farms located in different voivodeships.

The questionnaire was composed of 27 questions divided into two groups. The first included the basic characteristics of the farms (gender, age, education, farm size, farming experience, farming type). The second part was mostly composed of questions on the barriers to the development of organic farming. The following groups of barriers were identified in the survey questionnaire: production and economic barriers, market barriers, and institutional and regulatory barriers. The farmers were asked to assess the barriers on a Likert scale going from 1 (not important) to 5 (very important) in order to express their opinion on each barrier's impact on the development of organic farming. The second part of the survey questionnaire also included questions indirectly related to barriers to organic farming, and was about (1) motives behind going organic, (2) assessing the production risk, (3) costs and benefits of organic farming, and (4) assessing the future development opportunities.

The three categories of barriers identified in the questionnaire (production and economic barriers, market barriers, and institutional and regulatory barriers) are partly underpinned by research carried out in other countries among farmers who discontinued their organic farming business. Before starting the survey, when preparing the research procedure and asking the research questions, an assumption was made that factors which contribute to making the farmers discontinue organic farming can be perceived by farmers currently engaged in organic farming as barriers to its development. Getting to know the farmers' opinions on these barriers could be a major source of information for decision makers in formulating an organic farming development policy designed to decelerate the crisis processes. The authors realize that some other barriers to the development of organic farming might exist, making the vast majority of (conventional) farmers disinterested in going organic. Exploring these farmers' opinions on the barriers to the development of organic farming is an important research problem, too. The three groups of barriers identified in this paper do not exhaust the list of obstacles to the development of organic farming. Indeed, the survey questionnaire does not include social and information-related barriers. At the end of the questionnaire, the respondents were given the opportunity to express their opinion and comment on other aspects of the study.

More than half of the interviewees were in the 31–54 age bracket (Table 1). Most interviewees had either a secondary (44.3%) or a tertiary (33.2%) education. Farmers with more than ten years' experience in running an organic farm had the largest share (60.1%). These were mainly farmers who

had established their holdings in 2006 or later. One in three farms had an area of 20–50 ha, one in five had an area of 10–20 ha, and one in ten had an area of over 100 ha. The group of organic holdings (both in the study sample and on a countywide basis) included more medium-sized (20–50 ha) and large farms (above 50 ha) than the group of conventional farms.

**Table 1.** Selected characteristics of farms covered by the survey.

| Identification | Levels of Variables | Percentage (%) | Absolute Frequency |
|---|---|---|---|
| Gender | Female | 27.4 | 72 |
| | Male | 72.6 | 190 |
| Age (years) | <30 | 2.6 | 7 |
| | 31–40 | 28.5 | 75 |
| | 41–50 | 51.2 | 134 |
| | 51–60 | 15.4 | 40 |
| | 61 and over | 2.3 | 6 |
| Education | Elementary school | 2.3 | 6 |
| | Vocational education | 20.2 | 53 |
| | Secondary education | 42.3 | 111 |
| | Higher education | 35.2 | 92 |
| Farm size (ha) | <5 | 11 | 29 |
| | 5–10 | 14.5 | 38 |
| | 11–20 | 22.5 | 59 |
| | 21–50 | 31.3 | 82 |
| | 51–100 | 11.5 | 30 |
| | 101 and over | 9.2 | 24 |
| Type of farming | Field crops | 27.8 | 73 |
| | Dairy cows | 18.7 | 49 |
| | Grazing livestock | 30.2 | 79 |
| | Mixed | 23.3 | 61 |

Source: own study based on survey data, $N = 262$.

The survey identified four agricultural types of farms: arable farms, dairy-cows (farms specializing in the rearing of dairy cattle), grazing livestock (farms specializing in the rearing of grass-fed animals), and mixed farms. One in three farms were grazing livestock holdings (30.2%), and the group of arable farms was nearly the same percentage (27.8%). The two other types of farming are mixed farms (23.3%) and dairy-cow farms (18.7%). Around 20% of farms covered by this study were engaged in a non-agricultural activity.

## 3. Results

At the beginning of the survey, the farmers were asked about their motives for going organic. The top three answers were economic reasons—access to financial support (63.8%), charging a higher price (59.4%) and making production more profitable (56.3%) (Table 2). The motives cited by more than a half of the farmers included quality improvements of food produced (52.4%). The next three replies had a similar percentage and involved health and environmental aspects: caring for one's family health (49.7%), improving soil fertility (49.2%), and caring for the environment (46.1%). The reduction in production costs (43.7%) and production in compliance with one's values (41.1%) were aspects of lesser importance, which, however, were indicated by a relatively large share of respondents. Conversely, improving animal welfare (32.4%) and lifestyle (28.9%) were of minor relevance.

**Table 2.** Motives behind going organic.

| Specification | Percent (%) |
|---|---|
| Accessing financial support | 63.8 |
| Selling at higher prices | 59.4 |
| Increased production profitability | 56.3 |
| Increased quality of food produced | 52.4 |
| Concern for the health of one's family | 49.7 |
| Improved soil fertility | 49.2 |
| Environmental concerns | 46.1 |
| Reduction of production costs | 43.7 |
| Production in compliance with one's values and beliefs | 41.1 |
| Improved animal welfare | 32.4 |
| Improved lifestyle | 28.9 |

Source: own study based on survey data, $N = 262$.

Opting for organic farming involves many changes to a farm, which contribute not only to an increase in certain benefits but also to an increase in costs. Farmers are interested in keeping their farms afloat in the long run if they subjectively believe that the increase in expected benefits is greater than the increase in costs, otherwise they will consider discontinuing organic production. The benefits of going organic cited by the farmers surveyed primarily included the economic advantages—increase in incomes supported by subsidies (77.1%), increase in profitability (63.4%), and high selling prices (54.2%) (Table 3). Farmers believe support for their farms is more important than production profitability, because a certain level of support can offset a decrease in productivity. The increase in incomes caused by an improvement in relationships between productive inputs was rated relatively low (43.1%). As regards costs, a large proportion of the farmers cited the decline in yields (78.9%), increased labor intensity (62.7%) and increased costs of compliance with organic production requirements (61.1%).

**Table 3.** Benefits and costs of organic farming as seen by farmers (% of replies).

| Benefits | Replies | Costs | Replies |
|---|---|---|---|
| Increase in incomes, supported by subsidies | 77.12 | Decline in yields | 78.94 |
| Increased profitability | 63.43 | Increased labor intensity | 62.71 |
| Higher selling prices than in conventional agriculture | 54.17 | Costs of compliance with organic farming requirements | 61.06 |
| Lower consumption of external inputs | 53.64 | Small (economically unviable) production scale | 47.32 |
| Increase in incomes resulting from an improvement in the relationships between productive inputs | 43.11 | High marketing costs | 45.63 |
| Increase in competitiveness | 43.11 | Dependence on export markets | 39.75 |
| Environmentally friendly production processes | 43.01 | High labor costs | 28.37 |

Source: own study based on survey data, $N = 262$.

The support policy is a major factor affecting farmers' decisions on whether to maintain or discontinue organic production. This is reflected in the distribution of the farmers' replies to the question on whether they would continue their organic production activity without financial support (Table 4). Seventy-one percent (71%) said they intended to discontinue organic production; this is especially true for two types of farming—grazing livestock farms (87%) and mixed farms (83%).

**Table 4.** Continued organic production in a scenario with no financial support.

| Farmers' Opinion | Total Sample (%) | Type of Farming | | | |
|---|---|---|---|---|---|
| | | Field Crops | Dairy Cows | Grazing Livestock | Mixed |
| Yes | 29 | 39 | 36 | 13 | 17 |
| No | 71 | 61 | 64 | 87 | 83 |

Source: own study, $N = 262$.

The largest share (31.1%) of farmers surveyed intend to maintain their current production levels; this is especially true for three types of farming—dairy-cow farms (35.8%), field crop farms (34.6%) and

grazing livestock farms (29.5%) (Table 5). The second most popular intention declared by the farmers is to decrease their production volumes (27.2%). This is of particular relevance to grazing livestock farms (29.4%) and mixed farms (28.9%). Note also that in these two types of farming the largest proportion of farmers say they intend to discontinue organic practices (19.1% and 22.4%, respectively). In the total sample, nearly one in five farms (18.3%) declare their readiness to discontinue.

**Table 5.** Plans regarding organic production within the next 5 years.

| Intent | Total Sample (%) | Type of Farming | | | |
|---|---|---|---|---|---|
| | | Field Crops | Dairy Cows | Grazing Livestock | Mixed |
| Increase production | 23.4 | 28.5 | 27.3 | 21.0 | 23.1 |
| Maintain the current production level | 31.1 | 34.6 | 35.8 | 29.5 | 25.6 |
| Decrease production | 27.2 | 24.8 | 21.2 | 29.4 | 28.9 |
| Discontinue and switch back to conventional farming | 18.3 | 12.1 | 11.7 | 19.1 | 22.4 |

Source: own study, *N* = 262.

A considerable proportion of farmers who say they intend to reduce organic production and are prepared to discontinue organic farming may mean that Poland will continue to witness a decline in the area and number of organic farms. Organic farmers face many types of risk (production, price and institutional risk) that add to the uncertainty of their business. According to 56.9% of respondents, organic farms struggle with greater risks than their conventional peers. In future, this opinion may affect the farmers' decisions on whether or not to continue organic production. The full conversion of an entire farm is riskier, and therefore half of respondents continue with non-organic production in parallel. This in turn requires them to skilfully use two farming systems in one farm and to pursue different goals. Moreover, the dual approach involves additional investments in farm restructuring.

The survey divided the barriers to the development of organic farming into three groups: production and economic barriers, market barriers, and institutional and regulatory barriers. The farmers assessed them on a Likert scale from 1 (not important) to 5 (very important). The farmers' opinions on production and economic barriers show that low yields (4.64), agri–technical barriers (4.37) and lack of access to organic seeds and fertilizers (4.35) are the most important aspects (Table 6). Low yields are a barrier of tremendous importance to all types of farming. This is related not only to organic farming methods but also to low-quality soils. Indeed, Polish organic farms have low-quality soils, which reduces their production capacity and has an effect on the structure of crops and animal numbers [55]. This may suggest that organic methods are chosen by farms that, due to natural limitations resulting from low-quality natural assets, have little room for improvement in their competitive position in conventional farming systems.

**Table 6.** Production and economic barriers to the development of organic farming.

| Specification | Level |
|---|---|
| Decline in yields | 4.64 |
| Agri–technical barriers | 4.37 |
| Availability of organic seeds and fertilizers | 4.35 |
| Maintaining soil fertility | 3.72 |
| High labor intensity | 3.46 |
| High production costs | 3.18 |
| Not enough processing plants | 2.34 |
| Availability of machinery and equipment | 2.12 |

Source: own study based on survey data, *N* = 262, from 1 (not important) to 5 (very important).

It seems that production barriers related to the structural characteristics of organic farms are decisive for their development capacity. One of the major challenges facing organic farming is to

minimize the decline in yields both at the conversion stage and after certification. This is also confirmed by replies to the question on how they perceive the production risk during the conversion period and thereafter (Table 7). In the conversion period, 52% of farmers viewed this risk as extremely high (vs. 34% after the conversion period). During a two- or three-year period, farmers incur costs involved in switching to organic methods, investment costs and costs of learning a new production method (a steep curve in learning costs), while not deriving any benefits from certification or from the price premium.

**Table 7.** Perception of the production risk depending on how long an organic farm has been active.

| Specification | Production Risk | | | |
| --- | --- | --- | --- | --- |
| | Extremely High | High | Low | Extremely Low |
| During the conversion period | 52.12 | 29.93 | 12.24 | 5.71 |
| After the conversion period | 34.30 | 25.14 | 23.17 | 17.39 |

Source: own study based on survey data, *N* = 262.

The second group of barriers to organic farming development identified in this study were market barriers. Paradoxically, despite the shortage in supply, many organic farmers failed to reinforce their competitive edge as the market evolved. The way the farmers see the market barriers suggests they struggle to sell their produce (Table 8). Three barriers indicated in the survey were ranked at the top (with an average rate of 4.5)—difficulties in selling (4.82), prices being too low in relation to costs (4.59), and insufficient demand (4.52).

**Table 8.** Market barriers to the development of organic farming.

| Specification | Level |
| --- | --- |
| Difficulties in selling | 4.82 |
| Prices being too low in relation to costs | 4.59 |
| Insufficient demand | 4.52 |
| A large number of intermediaries in the supply chain | 4.34 |
| High margins charged by intermediaries | 4.21 |
| Low market power of organic farms | 4.17 |
| Lack of local markets | 3.81 |
| Markets being geographically distant | 3.47 |
| Risk of a fall in prices of organic food | 2.63 |

Source: own study based on survey data, *N* = 262, from 1 (not important) to 5 (very important).

As regards institutional and regulatory barriers, the greatest importance is attached to restrictions involved in national-level policies, especially including frequent amendments to legal regulations on organic farming (4.8), amendments to the principles of eligibility for payments (4.6), legal vagueness (4.2), and requirements derived from production standards (3.9) and certification (3.4) (Table 9).

**Table 9.** Institutional and regulatory barriers to the development of organic farming.

| Specification | Level |
| --- | --- |
| Frequent amendments to legal regulations on organic farming | 4.82 |
| Frequent amendments to the principles of eligibility for payments | 4.60 |
| Legal vagueness | 4.21 |
| High organic production standards provided for in the regulations | 3.92 |
| Burdensome certification requirements | 3.46 |
| Complicated documentation requirements | 3.25 |
| Insufficient levels of financial support | 3.23 |
| Low levels of non-financial support | 3.04 |

Source: own study, *N* = 262, on a scale from 1 (not important) to 5 (very important).

In the final part of this survey the farmers were asked about the determinants of organic farms' development capacity (Table 10). They rated increase in production support very high on the Likert scale (4.53). The interviewees believe the increases in financial support (4.53), in production volumes (4.28) and in farm area (4.17) play a greater role in improving organic farming's development capacity than the integration with other operators (2.57). The opportunities brought by cooperation induced by integration with other operators were ranked moderately high by the farmers. This issue needs to be explored in greater detail and therefore will be discussed later on.

**Table 10.** Drivers of development opportunities of organic farms.

| Specification | Level |
|---|---|
| Increased support for organic farming | 4.53 |
| Increased production and supply | 4.28 |
| Specialization of organic farms | 4.17 |
| Increase in farm area | 3.92 |
| Simplification of formal requirements for certification and inspection | 3.43 |
| Stability of laws applicable to organic farming | 3.29 |
| Horizontal integration of organic farmers | 2.57 |
| Networking in the supply chain | 2.32 |

Source: own study based on survey data, *N* = 262, from 1 (not important) to 5 (very important).

## 4. Discussion

These findings have important research and practical implications for organic farming. The survey shows that Polish organic farm owners view economic motives (accessing financial support, selling at a higher price and making production more profitable) as being more important than non-economic factors (caring for one's family, improving soil fertility, caring for the environment, making production compliant with what one values and believes in, improving animal welfare). The authors' findings are partly corroborated by other studies [36] according to which the profile of organic farmers is undergoing changes. New farmers are more pragmatic and business-oriented. This study suggests that—due to the great importance of economic incentives—Polish organic farmers are dominated by a group referred to by Darnhof et al. [27] as "pragmatic organic" farmers. Just like the "pragmatic organic" group, Polish farmers are guided by the prospect of secure revenues, primarily supported by organic payments. However, as their revenues depend on subsidies, they are increasingly prone to political manipulation [61].

One of the major findings from this study is the farmers' opinion on whether their decision to continue organic production depends on support. In the absence of support, nearly three-quarters of the farmers surveyed would discontinue organic production. This is especially true for two farm types: grazing livestock farms and mixed farms with low production profitability ratios. This can be explained by the fact that the share of support in their incomes is relatively high (above what is experienced in other types of farming) [48]. It follows from this study that financial support for Polish organic farms has an impact on their incomes and depends on the type of farming and farm area [44,45]. Based on an analysis of the production and economic conditions of organic farms covered by the Polish FADN (Farm Accountancy Data Network) (over 300 farms), Juchniewicz and Nahtman [62] demonstrated that the viability of most of them depends on external support. Organic farms attain low levels of production efficiency or even incur production losses, as is the case for grazing livestock and mixed farms; for instance, in 2018, the share of subsidies in the incomes of these two types of family farm was 127.3% and 118.3%, respectively. In dairy-cow farms and arable farms, the ratio was 61% and 86%, respectively. This means that grazing livestock and mixed farms are unable to survive without support. Due to poor production efficiency, Poland has a relatively large proportion of farms engaged in a combination of organic and non-organic production activities. This secures them against a potential increase in the production risk, as it provides them with an opportunity to gradually expand or reduce their organic production activity depending on how profitable it is.

This study shows that more than half of the farmers view the organic production risk during the conversion period as very high. Although they believe it to be less severe subsequently, it still is perceived as relatively high. This may suggest that when making their decision to go organic, the farmers covered by this study were unaware of the related risks. It is difficult to compare these findings against a study based on another methodology which demonstrated that organic farmers are much more risk-averse than their non-organic peers [63–65].

This survey suggests that the production risk poses a major problem for organic farmers. However, further research is needed to determine the impact of the perceived production risk on farmers' future decisions on whether or not to continue organic production.

As an important finding, this study explored farmers' plans regarding organic production. The authors found that nearly one in five farms (18.3%) want to discontinue organic production. This is especially true for two types of farming: grazing livestock farms and mixed farms. Other studies have reported smaller percentages of farmers who declared a willingness to discontinue organic farming. For instance, this was true of 13% of farmers in a Danish survey [66]. In a study carried out in Austria, 13% of organic farmers planned to switch to conventional farming [67]. According to an Austrian research project three years later, 13% of farms had actually done so. However, only some of these were part of the group which had earlier declared their willingness to discontinue organic farming. While the intention to discontinue is not enough to provide a complete picture of the farmers' actual behavior, it is an important piece of information on how intense that process will be in future.

The farmers' declared intention to discontinue organic production may suggest that the Polish organic farming sector will continue to follow the downward trend in the years to come. This information is crucial to decision-makers, especially when determining the instruments and measures to be used in minimizing the number of farmers who quit the organic sector. The lack of activity in agricultural policy, viewed as an enabler, can perpetuate the disruptions affecting organic farming. This is all the more justified since Polish organic farming is an integral part of the emerging market for organic food, which clearly suffers from a strong mismatch between demand and supply.

There is a relationship between what farmers intend and the barriers to the development of organic farming. The group of market barriers is strictly related to production barriers, because farmers view low yields and limited access to fertilizers and seeds as the two top-ranked obstacles; this is largely the reason for low levels of supply. In turn, low yields and production volumes are the reason why many farmers see organic farming as being risky. This is of particular importance in the conversion period, when the farms' production processes lose their conventional characteristics while not yet being fully organic. Compensation for the costs of organic production methods should therefore be high enough to minimize the likelihood of a discontinuation of organic farming. The best agri–technical practices need to be implemented in order to reduce the risk of lower yields. Some opportunities for reducing the production barriers can be seen in innovative solutions regarding new sources of fertilizer and plant protection products and techniques. However, this depends on whether knowledge is developed and transferred to the organic farming sector.

In Poland, the agricultural policy implemented after 2004 resulted in a quantitative growth of organic farming (an increase in the number of farms and in the area of organic farmland), but it failed to trigger a market effect in the form of increased food supply. Production volumes are small, production sites are dispersed, and the farmers have a limited capacity to attain the expected price premium. Organic farms find it difficult to sell their products either directly on local markets or through modern distribution channels [68]. As regards sales in the local market for organic food, a niche market, farmers take costly, time-consuming marketing measures to achieve the most advantageous price premium, which, like the subsidies, determines the cost-efficiency of their production system. The weaknesses of local markets are the low, sporadic demand and low consumer purchasing power [69,70].

Organic food sales are mainly based on regular demand, which creates a perspective for the development of this market [71,72]. The competitive position of organic farms therefore currently

depends on sales through modern distribution channels, which play an increasingly important role in the organic food market. In the supply chain, the conditions of cooperation are set by supermarkets and hypermarkets rather than by farmers, especially as regards the size of batches ordered, margin and price levels, and payment terms. Farmers have a limited capacity to adapt to these conditions because the quantities they supply are very small. This is decisive for the farmers' small contribution to value added and for the price levels, which are formed by intermediaries to a greater extent than by agricultural producers. Although price levels in the organic food market are high, research shows that they fail to fully offset the costs, because the traders, not the farmers, are the ones who reap the market benefits in the form of high margins. Paradoxically, this is why farmers find the price levels of organic food to be excessively low, whereas consumers consider them to be high. According to some studies, the vast majority (91%) of organic farmers expect the prices of their products to be higher than those of conventional products [73].

Another finding from this study is the farmers' opinion on institutional and regulatory barriers affecting organic farming. The results indicate that these barriers are of crucial importance and are mainly related to amendments to organic farming regulations. Similar conclusions were drawn from an American study [74], which found regulatory issues to be the most frequent reason why unregistered organic farmers switched back to conventional farming. In that study, farmers who continued their organic business were more concerned about regulatory matters (laws, certification, record-keeping) than those who went back to conventional practices.

In Poland, the main institutional problem is the instability of laws applicable to organic farming, which adds to the farmers' uncertainty and decision-making risks. According to North, underpinning the economy with an adequate institutional framework is the only way to reduce uncertainty as an inherent part of the economic environment [75]. He believes the institutional structure to be a factor that can generate conditions which result in either efficiency or stagnation. His paper claims that market efficiency depends not only on the productivity but also on the quality of institutions. From the perspective of institutional conditions, Polish organic farms face many operational difficulties due to frequent amendments to legal and political principles. For instance, between 2014 and 2017 there were seven amendments to the 13 March 2015 regulation on the detailed conditions and procedure for granting financial aid under the "organic farming" measure covered by the 2014–2020 Rural Development Programme. This testifies to the poor efficiency of the Ministry of Agriculture and Rural Development, as the main game-setter for the organic farming sector and as the holder of instruments and financial resources used in stimulating its development.

In the initial period that followed the introduction of subsidies in Poland, many farmers decided to shift to organic methods despite having insufficient knowledge and limited access to information on the technical and production-related aspects of these practices. The growing interest in organic farming was supported by the government's policy, which, in a ten-year period starting in 2004, was focused on stimulating environmental actions by farmers. In that period, farmers were provided with organic payments without being required to couple production with the market or engage in a combination of crop and animal production. This policy resulted in a low supply of organic food, market imbalances, and unsustainable feed and fertilizer management at the microeconomic level. That period did not witness any market effects of the quantitative growth of organic farming. Neither did the farmers strengthen their market position. This is why the farmers surveyed found both market barriers and institutional and regulatory barriers to be highly relevant.

In 2014, institutional changes regarding the conditions for granting organic payments initiated a new stage in the development of organic farming, and resulted in a shift from quantitative changes (increase in the number of farms and farmland area) to qualitative changes (production growth and an effect on supply). The period that started in 2014 is a "creative destruction" phase, which, in the long run, should result in the reduction of existing barriers to the development of organic farming, and in positive production, market and environmental effects. Initially, the introduction of the requirement to couple organic production with the market resulted in a decline in the area and number of organic farms.

However, in the long run it should trigger an increase in production volumes and, as a consequence, strengthen the competitive edge of organic farmers. The authors' findings suggest that 2014 was a breakthrough which started the decline in organic farming, as farmers did not have enough time to prepare for changes in the requirements imposed under the 2014–2020 RDP. In Poland, the volatility of legal regulations applicable to organic farming became an institutional barrier, and made organic farmers increasingly view their business as risky.

## 5. Conclusions

The key recommendation for decision-makers, as derived from this study, is to develop a stable institutional framework for the development of organic farming. Amendments to agricultural policy should be foreseeable and implemented well in advance so farmers have enough time to adapt to the new operating conditions. New regulations should be preceded by an evaluation of feedback from farmers, and be supported with rigorous information and agricultural consultancy measures. Another essential aspect is to implement initiatives designed to improve the quality of regulations and simplify the administrative burden involved in organic farming.

It is also recommended to support organic farmers' efforts to establish network links of a different nature with various participants of the organic food sector. This is because the farmers need to enhance their knowledge and skills related to production technologies, but also to learn more about the conditions for operating in the market and how to deal with formal and legal procedures applicable to the organic farming sector.

Characteristically, Polish organic farmers demonstrate low participation rates in associations and organizations established to put political pressure on strengthening the changes that promote the development of organic farming. This can be explained by low levels of public confidence and a historical aversion to collective forms of collaboration. This study suggests that the farmers underestimate the benefits of collaborative integration, which is not only a way to achieve synergies in solving production and market problems but can also provide an opportunity to acquire and exchange information and enhance knowledge, which is highly needed in organic farms. In a context of production and market restrictions, horizontal and vertical collaborative networks improve the farmers' adaptive capacity. Their overarching purpose is to establish conditions for strengthening the farmers' competitive edge in relation to other market actors. In the future, the growing market presence of large wholesalers, intermediaries and retailers could pose a threat to organic farmers and undermine growth in their incomes. One of the ways to counteract these risks is by creating various forms of cooperation to preserve the identity and boost the development of organic farming. Therefore, it is recommended to support the organic farmers' efforts made to establish network links of a different nature with different participants of the organic food sector. This is because the farmers need to enhance their knowledge and skills relating to production technologies, but also learn more about the conditions for operating in the market and how to deal with the formal and legal procedures applicable to the organic farming sector.

This paper focused on the opinion of Polish organic farmers regarding the barriers to the development of organic farming. However, many of these findings are relevant to a broader debate on barriers to the development of organic farming in other countries. The findings regarding the role of institutional barriers and communications from regulatory institutions which affect the farmers' decision-making processes are of particular importance. They are a part of the institutional framework for state administration in the domain concerned. At the same time, they provide a specific indicator of governance quality, because they are directly linked to the implementation efficiency and stability of the policy adopted.

Getting to know the barriers to the development of organic farming is important in order to understand the reasons behind the decline in the area and number of organic farms in Poland. The purpose of this study was to enhance the knowledge on how the barriers to the development of organic farming are viewed by today's organic farmers. It can also have some practical implications

for a modification of the organic farming policy that prevents a further decline in this sector. The case study on Poland, which is among the emerging markets for organic food, shows that a stable and coherent support policy is a condition for organic farming development.

This study does not address the problem of barriers to organic farming development in an exhaustive manner. It is therefore important that such research should be continued in future, taking feedback from different players on the organic food market into consideration. This study used some quantitative techniques to retrieve the farmers' opinions on the barriers to the development of the organic farming sector. We believe the study could be extended with in-depth interviews with organic farmers and with those who discontinued their organic farming business. This would enable a better understanding of limitations to the development of organic farming.

**Author Contributions:** Conceptualization, W.Ł.; data curation, W.Ł.; methodology, W.Ł.; resources W.Ł. and S.K.; formal analysis, W.Ł.; investigation, W.Ł.; writing—original draft preparation, W.Ł. and S.K.; visualization, W.Ł. and S.K.; writing—review and editing, W.Ł. and S.K.; project administration, S.K.; funding acquisition, W.Ł. All authors have read and agreed to the published version of the manuscript.

**Funding:** This research received no external funding.

**Conflicts of Interest:** The authors declare no conflict of interest.

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
