# Peer review of "Barriers to the Development of Organic Farming: A Polish Case Study"

_agriculture, doi:10.3390/agriculture10110536_

Round 1
Reviewer 1 Report
Overall I find this study to be robust and contribute with new insights on an important topic. It is well written and structured. There are some aspects where I think the discussion could be deepened, to go further beyond the somewhat obvious conclusion regarding the importance of institutional support. I think that the authors definitely have capacity of doing this.
Background section – references are missing for much of this section.
I would also like to know some basics on what the organic market looks like in Poland; what are the conditions in terms of consumer demand for organic? Is it also fluctuating like this, or growing, declining…? Is this important in shaping the political commitment (or lack thereof?)
Line 127: number of farms out of how many in total?
Methodology – I would like to know more about how the questions for the questionnaire were developed. Were they (the categories and the individual questions within) based on past research on barriers to organic farmers (from other countries)? From where did the authors get the ‘types’ they mention? Since it was a survey and not an interview with open ended questions, I assume these were pre-defined categories, and for that reason it is important to know how these were defined (and what may have been missed, as a result of not asking about it!). Were there ways for respondents to add information/perspectives outside of the set questions?
Results
Motivation for going organic – were these also set options that the farmers could choose from, or was it an open ended question where the researchers later categorized the answers?
Point that starts on line 379: About farmers underestimating the role of collective integration. I agree with these points and think they are very important (see point on discussion for example – not only the economic aspects you mention, but also politically) but I don’t think it fits in the results section. It requires references to support the points you make – that farmers “underestimate” the importance of this, and why. I recommend more simply stating that farmers did not rank this very highly, and indicate that that this is noteworthy and you will return to it in the discussion.
Discussion
Somethings need to be added to the discussion but it is already quite long. I don’t see any particular parts as redundant, so would seek to write more concisely in order to address the below comments without adding too much.
One thing that I think should be mentioned and discussed as a limitation either here or in the methodology section is that there may be other barriers which are not seen because the research targets farmers who already engage in organic production. There may be other barriers which would emerge if the research would (also) target farmers who have NOT opted to go organic. This does not weaken the study but is a limitation that should be acknowledged.
What I mainly miss is a deeper discussion about what the research suggests is missing (stable institutional support for organic farming). The study raised questions about the deeper reasons for why it has been so poor/unstable, and what is needed to address this. I think it is quite well established already that institutional support is important for organic farming to grow and be sustained, so this is not very surprising; but are there insights from other countries that can be used, in dialogue with your findings, to further a discussion on this? For example, to what degree are Polish organic farmers organized in ways that can enable them to have political influence and call for reforms that better support organic farming? Can the finding here (that farmers don’t seem to see this as very important) be further discussed in relation to this? Perhaps it is partly linked to the characteristic of being “pragmatic” organic growers – unlike some countries, it didn’t emerge out of a collective (often environmentally-oriented) movement, but individual farmers seeking opportunities.
Author Response
Authors_responces_1

Reviewer 2 Report
I would like to thank the authors for their interesting contribution to the organic farming debate with Polish perspective and focus on the barriers to organic farming. The study is well written, clear, important and engaging.
Below you will find some of my suggestions on how to make it even more clear.
Introduction
The introduction section is very extensive and covers a plethora of research. Could you please formulate the objectives of your study in the last paragraph of the introduction? They are somewhat visible, but not clear enough.
47-48 The discontinuation rate for 2005 seems quite outdated. Could you find some more recent data?
48-50 The conclusion that the discontinuation of organic farming practices means that organic farming faces some microeconomic barriers is a bit far-fetched. Perhaps it would be better to say may imply instead of means.
54-61 I am wondering if it is needed to cover the studies dealing with the factors for conversion to organic farming in the introduction, since your study is focusing on the barriers, and there is already a lot of information in the introduction.
Background
In my opinion, this part is too extensive and relies too much on the subjective interpretation of the trends by the authors. Also, there are very few references. The numbers of farms and area under organic production and the changes also lack references. I see a reference under the Figure 1, but it also needs to be included in the text. I am wondering if this extensive background section could be compressed to a couple of paragraphs and added to the introduction section? There is a lot of information in here that is redundant, subjective and doesn’t really add to the flow. Less is often more. This is my opinion, but even if you decide to keep it, please reduce this section substantially.
155-161 I find these statements very arguable. Environmental sustainability is not achieved through animal husbandry at farms, nor is it a prerequisite for organic farming, as implied by authors. The absence of livestock on the farms does not contradict the concept of environmental sustainability, perhaps only to some degree makes farms rely more on outsourcing of the organic fertilizer. There are, in fact, plenty of environmental issues connected to animal rearing and livestock farming is in many ways the greatest threat to environmental sustainability. Furthermore, farms commercial capacity does not have to be endangered by not supplying livestock products to the market. If the farm decides to focus on another branch of products, it can be potentially even more successful, due to the fact that livestock farming is very cost intensive. Perhaps that was part of the reason for Polish organic farmers opting out of livestock farming. I suggest removing this part, as it is very much opinion-based and not fact-based. Or if you find a good reference for these claims, please use it.
165-169 This part is also somewhat opinion-based. There could be other forces behind the fact that the farmers opted out of the organic movement, not just the changes in the programme. Perhaps they were losing profits rapidly and didn’t feel sufficiently compensated by the market… How do you know that these farmers were poorly linked to the organic food markets? Please provide a quote for this statement.
193-215 Is the detailed composition of organic farms in Poland really relevant for the article? It seems to be overly extensive and I think this part could be significantly reduced or removed.
Results
This section is comprehensive and very well presented.
379-388 This paragraph is an interesting discussion of the results and should therefore be moved into the discussion section.
Discussion and conclusions
This part is very comprehensive, engaging and interestingly written. I would only suggest to split it into two sections- Discussion and Conclusions with implications for policy makers (probably from line 527 onwards).
Author Response
Authors_responces_2
